# Performance and Thermal Perceptions of Runners Competing in the London Marathon: Impact of Environmental Conditions

**DOI:** 10.3390/ijerph18168424

**Published:** 2021-08-10

**Authors:** Tim Vernon, Alan Ruddock, Maxine Gregory

**Affiliations:** Sport and Physical Activity Research Centre, Department of Sport and Physical Activity, Sheffield Hallam University, Sheffield S10 2BP, UK; a.ruddock@shu.ac.uk (A.R.); m.gregory@shu.ac.uk (M.G.)

**Keywords:** running, endurance, heat, thermoregulation, weather, cooling

## Abstract

The 2018 Virgin Money London Marathon (2018 VMLM) was the hottest in the race’s 37-year history. The aims of this research were to (1) survey novice mass participation marathoners to examine the perceptual thermal demands of this extreme weather event and (2) investigate the effect of the air temperature on finish times. A mixed-methods design involving the collection of survey data (*n* = 364; male = 63, female = 294) and secondary analysis of environmental and marathon performance (676,456 finishers) between 2001 and 2019 was used. The 2018 VMLM mean finishing time was slower than the mean of all other London marathons; there were positive correlations between maximum race day temperature and finish time for mass-start participants, and the difference in maximum race day temperature and mean maximum daily temperature for the 60 days before the London Marathon (*p* < 0.05). Of the surveyed participants, 23% classified their thermal sensation as ‘warm’, ‘hot’ or ‘very hot’ and 68% ‘thermally comfortable’ during training, compared with a peak of 95% feeling ‘warm’, ‘hot’ or ‘very hot’ and 77% ‘uncomfortable’ or ‘very uncomfortable’ during the 2018VMLM. Organisers should use temperature forecasting and plan countermeasures such as adjusting the start time of the event to avoid high temperatures, help runners predict finish time and adjust pacing strategies accordingly and provide safety recommendations for participants at high-risk time points as well as cooling strategies.

## 1. Introduction

Over one million runners participate in marathons annually [1], with the Virgin Money London Marathon (VMLM) attracting around 40,000 participants. The demands of marathon running are considerable irrespective of performance standard; the energy expenditure of female and male runners finishing between 2 and 4 h is within the region of 2000 to 2800 kcal, placing extensive strain on metabolic, cardiorespiratory, thermophysiological, mechanical and perceptual regulatory systems [2,3]. Such demands are intensified in hot and humid conditions where finishing times are impaired [4,5,6], and in-race withdrawals increase, particularly when air temperature exceeds 20 °C [6].

Substantial evidence supports the increased likelihood of extreme global weather events in the forthcoming decades, including periods of unseasonably cold and hot weather. In February and March 2018, the UK experienced two severe winter weather events with unseasonably low temperatures and significant snowfall [7]. For runners preparing for the VMLM, these low temperatures posed logistical training demands (icy roads) and altered perceptual and physiological responses, potentially impairing preparation for a spring marathon. These weather events were followed by unseasonably high temperatures between the 18th and 22nd of April, including the UK’s warmest April day since 1949 at 29.1 °C. This coincided with the hottest London Marathon on record of 24.1 °C that took place on Sunday 22nd April 2018 (10 am start), with many mass participation runners still on course when this temperature was recorded. These data, however, do not capture the potential environmental demands within the microclimate on course, whereby runners grouped nearby can experience increases in mean radiant temperature of 2 °C and humidity, coupled with decreases in radiative and convective heat transfer, increasing autonomic thermoregulatory challenges [8]. It is unclear how thermal sensation and comfort, which are integral to behavioural thermoregulation and dominant factors that dictate perceived exertion in hot conditions [9], are influenced in these specific environments. Furthermore, in the final weeks and days of marathon preparation, runners were faced with two conflicting extreme weather events. In the future, extreme weather events are likely to increase in either frequency and/or severity, posing risks to the short- and long-term health and wellbeing of marathon runners.

As such, the aims of this research were to (1) identify historical temperature data for all London Marathons to place the extreme weather events of 2018 in context and (2) survey a sample of novice mass participation marathoners who started the 2018 VMLM to examine the thermal demands of the extreme weather event on race day. This investigation is the first in the context of the London marathon and warranted on the basis of (1) the addition to our knowledge and understanding of perceptual demands of mass participation runners in the Virgin Money London Marathon and (2) identification of potential areas whereby participants and race organisers might seek to change their practice in preparation for, or on the day of, the event when high temperatures are forecast.

## 2. Materials and Methods

### 2.1. Research Design

A mixed-methods design involving the collection of survey data and the analysis of environmental and marathon performance data was used to examine the experiences of and the effects of air temperature on participants running a marathon in the heat. The local ethics committee approved the study (ER6896994). All participants provided digital informed consent, and the investigation was conducted in accordance with the Declaration of Helsinki (7th revision).

### 2.2. Participants

The finish times of 676,456 finishers (male, 450,071; female, 226,385) of the London Marathon from 2001 to 2019 were extracted from the official website of the VMLM [10] and the marathon archives website [11]. This included the 40,179 participants who completed the 2018 VMLM (male, 23,701; female, 16,478).

A random sample of participants (*n* = 364; male = 63, female = 294; age = 41.4 ± 8.3 years; mass = 72.2 ± 19.9 kg; stature = 168.6 ± 9.5 cm) from the 2018 VMLM (subsequently referred to as survey participants) completed an online survey relating to their expectations and experiences of the event, including expected and actual finish time, perception of temperature and thermal comfort during the marathon and during training over the winter months. The survey was distributed 7 days post-event via social media platforms and hosted on the Key Survey platform (www.keysurvey.co.uk—accessed 1 April 2018). The group was mostly inexperienced, with the 2018 VMLM being the first marathon for 63% of runners; 33% of runners had participated in between two and five marathons and only 4% had run more than six marathons.

Participants were not invited to comment on the study design; however, they were consulted during the writing of a plain language summary for dissemination to their peers and distribution to participant groups.

### 2.3. Data Analysis

Hourly temperature data (°C) were acquired from the UK’s National Meteorological Service (the Met Office) for the date of the London Marathon and the preceding 60 days for the years 1981 to 2019. These data were recorded as per the Met Office standards at a meteorological station located in St James’s Park London, chosen for its central location on the London Marathon route. Data were processed, plotted and analysed alongside marathon performance data using customised Microsoft Excel Software (Microsoft Corp, 2013) and SPSS (Version 24.0. IBM Corp, Armonk, NY, USA) and assumptions for parametric statistical analyses were performed. Pearson product-moment correlation was used to investigate relationships between maximum race day air temperature and the differential temperature between race day temperature and average maximum temperature for the previous 60 days (race day temp–average max temp over previous 60 days) on the average London Marathon finish time between 2001 and 2019. Thresholds of 0.1, 0.3 and 0.5 for small, moderate and large correlations [12] and 0.7 and 0.9 for very large and extremely large correlations [13] were used to interpret relationships between variables. Independent t-tests were performed to assess the difference in London Marathon finish times, air temperature and expected versus actual finish time from survey respondents. Statistical significance was set at *p* < 0.05. Thermal sensation [14] and comfort data [15] were analysed using a Friedman test and Dunn-Bonferroni post hoc test, and statistical significance was set at *p* < 0.05.

## 3. Results

### 3.1. Environmental Data

The maximum air temperature recorded by the Met Office on the day of the 2018 VMLM was 24.1 °C, hotter than the mean maximum race day temperature for every other year (15.2 ± 3.3 °C). The maximum daily temperature in the 60 days before the 2018 race day was also lower than previous years (11.3 ± 6.3 °C vs. 12.1 ± 1.6 °C), demonstrated in Figure 1, which also shows the three extreme weather events of 2018 (a, b and c).

### 3.2. Finish Time

The total field (*n* = 40,179) mean completion time for the VMLM 2018 was 20 min slower than the mean of all other years between 2001 and 2019 (290 ± 55.5 min vs. 270 ± 64.1 min). There was a very large positive correlation (r = 0.89, *p* < 0.05) between maximum race day temperature and mean completion time for mass participation runners (*n* = 676,456) (Figure 2a), and a very large positive correlation (r = 0.85, *p* < 0.05) between the differential between race day temperature and mean maximum daily temperature for the 60 days before the London Marathon and mean completion time for mass participation runners (Figure 2b).

The mean finish time of survey participants for the 2018 VMLM (Figure 3) was 337 ± 51 min, slower than the overall mean finish time for the 2018 VMLM of 290 ± 64 min and 47 ± 30 min or 14% slower than the survey participants reported estimated time (*p* < 0.05).

For those who had run marathons previously, the mean finish time for the 2018 VMLM was 40 ± 27 min slower than their previous best marathon time (317 ± 71 min vs. 278 ± 60 min; *p* < 0.05). Out of 101 runners who had previously run a marathon, one runner ran faster in 2018 VMLM by approximately 1 min.

### 3.3. Perception of Heat

To assess thermal sensation, an eight-point scale (very cold (1) to very hot (8), ‘very hot’ added to the original ASHRAE 2017 scale) was used to ask participants, ‘How hot did you feel during the marathon, and how does this compare with your training?’ They reported feeling hotter (*p* < 0.01) during the 2018 VMLM (mean = 6.8 ± 1.1, median = 7) compared with a typical training run (mean = 4.2 ± 1.7, median = 4) (Figure 4). Runners felt hotter as the race progressed (0 to 10 km compared with all sections, *p* < 0.01) (0 to 10 km mean = 6.2 ± 1.2, median = 6; 11 to 20 km mean = 6.9 ± 1.0, median = 7; 21 to 30 km mean = 7.2 ± 0.9, median = 7; 31 to 42 km mean = 7.0 ± 1, median = 7), with 76% feeling ‘warm’ to ‘very hot’ in the first 10 km, increasing to 91% of participants after the first 10 km. The percentage of participants who felt ‘warm’ to ‘very hot’ during a typical training run was 23%.

Runners felt more comfortable with their body temperature during training (mean = 3.5 ± 0.9, median = 4) compared with all sections of the 2018 VMLM (0 to 10 km mean = 2.8 ± 1.0, median = 3; 11 to 20 km mean = 2.3 ± 0.9, median = 2; 21 to 30 km mean = 1.9 ± 0.9, median = 2; 31 to 42 km mean = 2.0 ± 0.9; median = 2). Runners felt increasingly more uncomfortable as the race progressed (0 to 10 km compared with all other sections, *p* < 0.05); however, there was no significant change in thermal comfort from 21 to 42 km). The percentage of runners feeling ‘uncomfortable’ or ‘very uncomfortable’ increased from 37% during the first 10 km to 77% between kilometres 21 and 30 (Figure 5). The majority of participants felt ‘comfortable’ during a typical training run; 11% of respondents felt ‘uncomfortable’ or ‘very uncomfortable’ with the temperature during a typical training run.

## 4. Discussion

To our knowledge, this investigation is the first to analyse historical weather data of the London Marathon in the context of previous weather data, assess thermoperceptual demands and determine the effect of ambient temperature on the finish time of runners. The findings from this investigation are that: (1) The 2018 VMLM was hotter than the mean of all other London Marathons. (2) The 2018 VMLM mean finishing time was slower than the mean of all other London Marathons. (3) In accordance with the aforementioned major findings, we found a positive correlation between maximum race day temperature and finish time for mass participants, whereby a hotter temperature was related to a slower finish time. (4) We also found a positive correlation between the difference in maximum race day temperature and the mean maximum daily temperature for the 60 days prior to the London Marathon, whereby a hotter race day temperature compared with mean training temperature resulted in a slower marathon finish time. (5) Only 23% of surveyed participants classified their thermal sensation as ‘warm’, ‘hot’ or ‘very hot’ during their training period compared with a peak of 95% of participants feeling ‘warm’, ‘hot’ or ‘very hot’ during the 2018 VMLM. (6) A total of 68% of surveyed participants felt ‘comfortable’ with their body temperature during training, whereas a peak of 77% felt ‘uncomfortable’ or ‘very uncomfortable’ during the 2018 VMLM.

Our findings that the 2018 VMLM was hotter and the finish time slower than previous London Marathons are in accordance with previous research indicating that finish times are slower in hot conditions compared with cooler conditions [4,5,6]. Predictions of performance impairment based on ambient temperature suggest a decrease in finish time between 1.5 and 2% for every 5 °C increase in ambient temperature above 10–12 °C [16,17]. Our data, which relate specifically to our sample from the London Marathon with mean finish times between 260 and 290 min, suggest there is approximately a 2.8% decrease in finish time for every 5 °C increase in ambient temperature above 12 °C. An approximate 2.6% decrease in finish time for every 5 °C increase in temperature differential is also evident when considering the difference between maximum race day temperature and the mean temperature 60 days before the London Marathon. This knowledge is important because it could be used before any London marathon to adjust pacing strategies by taking into account the St James Park weather forecast. For example, a runner with an expected finish time of 3 h 59 min derived from training data before the London Marathon at a mean temperature of 12 °C would need to adjust their performance time by approximately 5.6%, from a mean pace of 5:40 min/km to 5:59 min/km if the expected maximum daily temperature at St James Park was forecast to be 22 °C.

Based on the 2018 VMLM temperature data, the expected performance decrement was approximately 6.6%; however, our sample of runners experienced a 14% decrement in finish time compared with their estimated finish time. Although we are not able to confirm the reasons for this difference, one explanation for this discrepancy might be accounted for by the pacing strategies employed by respondents who attempted to start the race at their expected race pace rather than adjust their pace to offset the impact of higher-than-expected temperatures on thermophysiological, energetic and perceptual demands. This discrepancy may be further accounted for by increases in microclimate temperature, in particular an increase in relative humidity caused by evaporative heat loss through sweating and respiration in our sample of runners, who were likely running together in tight groups with limited airflow. Indeed, 101 runners in the present sample ran slower than their previous marathon best, indicating that the environmental conditions in the lead-up to the marathon and on the day of the event influenced performance to a large extent. However, it is also likely that the expected finish time of runners would be faster than their actual completion time even in typical temperatures, given that most runners in this sample were novices and runners, in general, are optimistic about their performances before a race. The complex interactions between environmental conditions, pacing strategies and runners’ previous experience likely account for this 14% discrepancy. This discrepancy highlights the importance of monitoring training data in the weeks before a marathon to accurately predict finish time before taking into account the impact of weather on health, wellbeing and performance.

Only 23% of surveyed participants classified their thermal sensation as ‘warm’, ‘hot’ or ‘very hot’, respectively, during their training period compared with a peak of 95% of participants feeling ‘warm’, ‘hot’ or ‘very hot’ during the 2018 VMLM. Moreover, 68% of surveyed participants felt their thermal comfort during training was ‘comfortable’, whereas a peak of 77% felt ‘uncomfortable’ and ‘very uncomfortable’ during the 2018 VMLM. During exercise in hot environments, the two key inputs directly related to the rating of perceived exertion (RPE) are the rate of increase in and/or magnitudes of thermal sensation, thermal comfort and cardiovascular strain [9]. Initial predictions regarding the intensity of exercise are primarily made based upon skin temperature, which has a large influence on ratings of thermal comfort and thermal sensation, followed by cardiovascular strain and ventilatory rate (breathlessness). It is possible that at the start of the marathon, when ratings of thermal sensation and comfort were more similar to training, RPE and, thus, pacing were also the same. Ratings of thermal sensation and comfort are important from both thermoregulatory and homeostatic perspectives as increases in both likely reflect an increase in whole-body thermophysiological demand, in particular body temperature and significant cardiovascular challenges to maintain homeostasis. Approximately 70% of respondents felt ‘hot’ from 11–20 km, peaking at 80% between 21–30 km, compared with 40% in the first 10 km. We do not have medical records accompanying these data; however, previous research conducted on the 2007 London Marathon, which was also relatively hot (air temperature = 19.1 °C) compared with previous years (air temperature = 11.6 °C), reported a mean finish time 17 min slower than previous years. In the 2007 London Marathon, there were 5032 runners treated by St John’s Ambulance, 73 hospitalisations, 6 cases of severe electrolyte imbalance and 1 death (hyponatraemia) compared with the 2008 London Marathon (air temperature = 9.9 °C) in which the number of runners treated was 4000. It is not possible to isolate heat-related issues within these figures [18]; however, there is an association between the percentage of withdrawals from races and increasing air temperature over 15 °C [6]. Given the several risk factors associated with exercise in hot environments [19], global recommendations for event cancellation based on Wet Bulb Globe Temperature (WBGT) should be considered as a guide [20]. The American College of Sports Medicine [21] and Roberts [18] suggest that non-elite runners should be closely monitored, and event organisers should also consider the cancellation of the event if the WBGT is within the range of 18.4 °C and 22.2 °C. These recommendations, however, do not take into account the acclimation state of runners and given the unseasonably low temperatures preceding the 2018 VMLM, it is reasonable to assume that the majority of mass participation runners, who trained in the UK, would not have had the opportunity to prepare in environmental conditions similar to race day. Indeed, our analysis suggests a strong relationship between the difference in maximum race day temperature and the mean maximum daily temperature for the 60 days before the London Marathon (Figure 2b).

### 4.1. General Practical Recommendations

Organisers of marathons, especially those in the United Kingdom and Northern Europe, should follow existing guidance on exercise in hot environments, particularly regarding fluid balance and provision of practical cooling strategies [20,22]. Specifically, organisers should (1) keep detailed logs of environmental temperature in the preparatory phases leading up to the event, use temperature forecasting and plan countermeasures such as adjusting the start time of the event to avoid high temperatures; (2) help runners predict finish time and adjust pacing strategies accordingly using temperature-based data analysis; and (3) use data regarding thermal perception to provide recommendations for participants regarding high-risk time points of events to help alleviate discomfort.

### 4.2. Limitations

The demographics of the survey respondents are not a fully representative sample of those participating in the marathon, as our sample were predominantly female, of middle age and slower than the average runner. The survey was released 7 days after the marathon, increasing the reliance on respondents to accurately recall events from race day. The weather data measured at St James’s Park, London, was chosen for its central location but may not be representative of the weather along the entire course. Similarly, while it is appreciated that not all marathon participants would have prepared for the marathon in or around London, the use of this meteorological site to quantify the extreme cold events in Spring 2018 allowed direct comparison with race day temperatures on the marathon course. The extreme weather events experienced in London and captured in the data from the St James’s Park London station in the lead-up to the 2018 VMLM are used as a reflection of the conditions that were experienced across the rest of the UK and large parts of Europe. While we are able to conclude that a runner’s pace slows with increased race day temperature, we are unable to identify whether this is a conscious decision or a subconscious response to the increased physiological demand placed on them by the high temperature.

Future research should aim to capture the responses of a more representative sample of runners, initiate and capture data earlier and collect primary data during the marathon using wearable sensors and multiple environmental condition sensors along the course.

## 5. Conclusions

The 2018 VMLM was hotter than previous marathons; this subsequently slowed runners finishing time and made runners feel ‘very hot’ and ‘uncomfortable’ for the majority of the race. Our findings have several practical implications, most notably the utilisation of a specific analysis on predicted race day temperature and expected finish time, as well as the identification of time points of the race that coincide with increased thermal demand. Integrating a range of strategies highlighted as a result of this research could help make the race safer and more enjoyable for mass participation runners.

## Figures and Tables

**Figure 1 ijerph-18-08424-f001:**
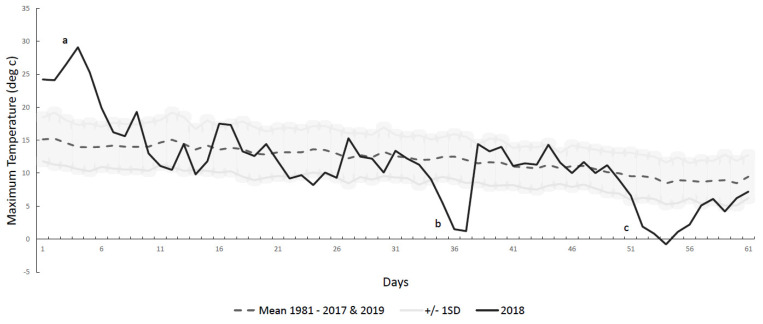
Maximum daily air temperature (°C) in the 60 days leading up to London Marathon race day for 2018 and mean ± SD across all other years 1981–2017 and 2019.

**Figure 2 ijerph-18-08424-f002:**
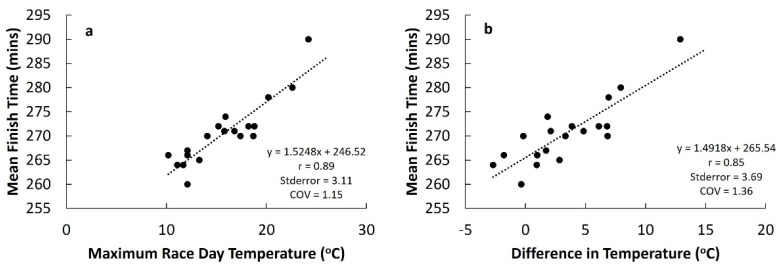
(**a**) Relationship between maximum race day temperature and mean finish time in the London Marathon 2001 to 2019. (**b**) Relationship between the difference in maximum race day temperature and maximum daily temperature during the 60 days prior to race day on mass participation and mean finish time in the London Marathon 2001 to 2019; stderror, standard error of the estimate; COV, coefficient of variation.

**Figure 3 ijerph-18-08424-f003:**
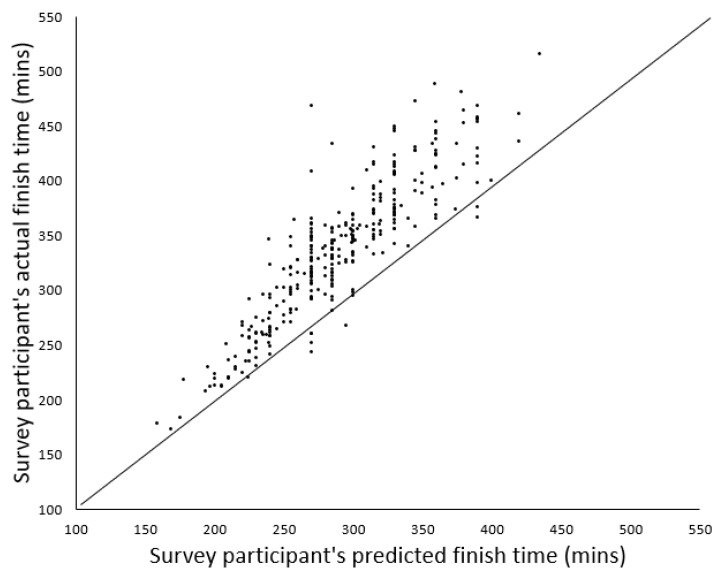
Estimated versus actual finish times in the 2018 VMLM for survey participants.

**Figure 4 ijerph-18-08424-f004:**
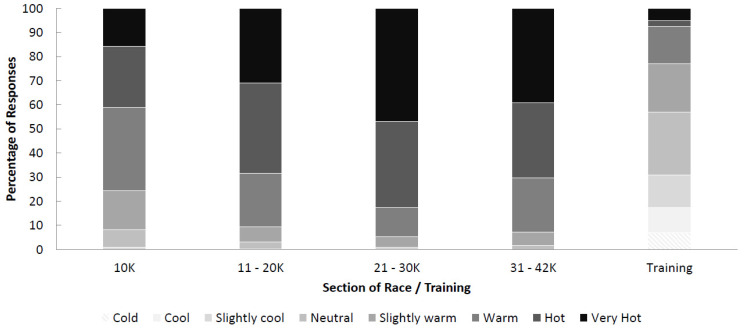
Thermal sensation during the VMLM 2018 and a typical training session during winter/spring in the UK in preparation for the 2018 VMLM.

**Figure 5 ijerph-18-08424-f005:**
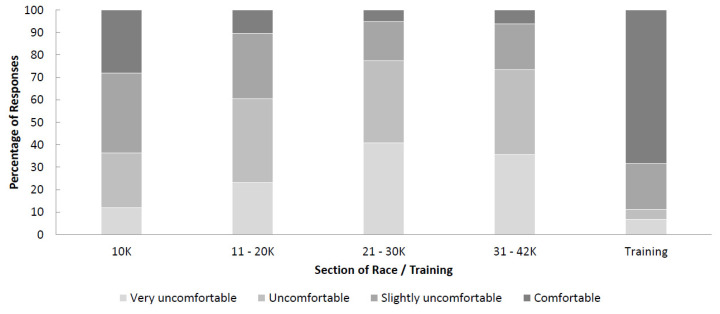
Thermal comfort during 2018 VMLM and a typical training session during winter/spring in the UK in preparation for this event.

## Data Availability

All meteorological and race data are available online at the websites of the Met Office and the VMLM. Aggregated and anonymised survey data are available on request.

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
