# Peer review of "Performance and Thermal Perceptions of Runners Competing in the London Marathon: Impact of Environmental Conditions"

_ijerph, 2021, doi:10.3390/ijerph18168424_

Round 1

Reviewer 1 Report

This study aims to: 1) identify historical temperature data for all previous London Marathons to place the extreme weather events of 2018 in context; 2) survey a sample of novice-mass participation marathoners to examine the thermal demands on race day and 3) investigate the effect of the extreme weather events on finish time.

Personally, I found this study very interesting because it could provide useful information to coaches, athletes and it could be very important from an organizational point of view. The findings have useful practical implications, especially concerning predictive actions and analysis about finish time and time-points with important thermal demands during the race. In addition, it seems to be the first investigation regarding the London Marathon taking into consideration the topic carried out in this manuscript. However, after reading and analysing it in more detail, I have some comments for improving even more the quality of this article.

INTRODUCTION

Line 35: Errors with non-separated words.

Lines 44-45: The degree symbol in number is located in a wrong way.

Line 50: The blank space before “It is unclear…”

Lines 52-53: Reiterative word “environments”.

MATERIALS AND METHODS

Line 71: Could you specify each design for the primary and secondary analysis? The first one is related to the data collection through a survey and the second one is a retrospective analysis, so for this reason even if you defined the design as “mixed-methods”, I think it would be better to add this information.

Line 79-80: The sentence is confusing for me. In this case, do you mean 180 male “winners” or “medallist” from London Marathon championship 2009-2019? Please clarify this point in a better way.

Line 84: Height is a more commonly used word.

Lines 78-95 & 100-102: I guess that the year range for retrospective analysis, the finish times and runners were selected based on the data availability but, were there any other factors for selecting these year ranges? To be honest, it was a bit confusing to read so many different year ranges.

Line 117: The information concerning SPSS it is reiterative. Please remove it.

DISCUSSION (major revision)

Even though the development of this section is readable and the authors offer arguments, I believe the description of the results is too detailed for a discussion since they have been already presented in the previous section. In addition, there is not enough discussion around previous studies on the topic, based the scarce number of references provided in the manuscript and discussed in this section. The arguments provided by the authors seem to be adequate but it would be necessary to shorten the amount of information purely related to results and go deeper into a proper study discussion by confronting with data and arguments from other studies.

LIMITATIONS

350-352: Although it is clearly a limitation, but with the tracking of so many previous years this is considerably minimised.

352-353 Introduce this information in the methodology rather than in limitations.

362-365 Very important for future research. To include questions in the survey concerning that issue specifically would be necessary.  

REFERENCES

Please, check all the references because the styles are mixed. You should report this section in Vancouver style.

Thank you very much for your consideration in reviewing this manuscript.

I hope to be helpful again on future occasions.

Author Response

Response to Reviewer One

This study aims to: 1) identify historical temperature data for all previous London Marathons to place the extreme weather events of 2018 in context; 2) survey a sample of novice-mass participation marathoners to examine the thermal demands on race day and 3) investigate the effect of the extreme weather events on finish time.

Personally, I found this study very interesting because it could provide useful information to coaches, athletes and it could be very important from an organizational point of view. The findings have useful practical implications, especially concerning predictive actions and analysis about finish time and time-points with important thermal demands during the race. In addition, it seems to be the first investigation regarding the London Marathon taking into consideration the topic carried out in this manuscript. However, after reading and analysing it in more detail, I have some comments for improving even more the quality of this article.

  • Thank you for taking the time to review the manuscript and for your suggestions, please find below our response and the changes that we have made to the manuscript as a result.

INTRODUCTION

Line 35: Errors with non-separated words.

  • Corrected

Lines 44-45: The degree symbol in number is located in a wrong way.

  • Corrected

Line 50: The blank space before “It is unclear…”

  • Corrected

Lines 52-53: Reiterative word “environments”.

  • Changed

MATERIALS AND METHODS

Line 71: Could you specify each design for the primary and secondary analysis? The first one is related to the data collection through a survey and the second one is a retrospective analysis, so for this reason even if you defined the design as “mixed-methods”, I think it would be better to add this information.

  • We have removed the reference to secondary analysis, the research design is an overview and data analysis is explained more thoroughly in section 2.3

Line 79-80: The sentence is confusing for me. In this case, do you mean 180 male “winners” or “medallist” from London Marathon championship 2009-2019? Please clarify this point in a better way.  

  • I have moved the definition of a “Championship Runner” from further down this paragraph.

Line 84: Height is a more commonly used word.

  • Stature is used as this is the specific noun for the natural height of a person, height can refer to the distance from the base to the top of any object.

Lines 78-95 & 100-102: I guess that the year range for retrospective analysis, the finish times and runners were selected based on the data availability but, were there any other factors for selecting these year ranges? To be honest, it was a bit confusing to read so many different year ranges.

  • Correct, the date ranges were based on the available data, we took everything that was available to increase the power of the study and help with the interpretation of our findings.

Line 117: The information concerning SPSS it is reiterative. Please remove it.

  • Removed

DISCUSSION (major revision)

Even though the development of this section is readable and the authors offer arguments, I believe the description of the results is too detailed for a discussion since they have been already presented in the previous section. In addition, there is not enough discussion around previous studies on the topic, based the scarce number of references provided in the manuscript and discussed in this section. The arguments provided by the authors seem to be adequate but it would be necessary to shorten the amount of information purely related to results and go deeper into a proper study discussion by confronting with data and arguments from other studies.

  • In the discussion we further interpret the results of our analysis and place this in context to findings our previous studies. Specifically, we highlight comparisons between performance impairment and previous studies and indicate the practical importance of our findings. We also discuss the potential reasons for the discrepancies between predicted marathon performance and actual marathon performance and the possible perceptual mechanisms that might play a key role in explaining these differences. Furthermore, we place our findings in context to previous studies that have suggested safe temperatures for marathon running and offer our opinion on how this might be influenced by environmental temperatures leading up to the event.

LIMITATIONS

350-352: Although it is clearly a limitation, but with the tracking of so many previous years this is considerably minimised.

  • This limitation doesn’t affect the thermal or performance data, only the survey data that we collected in 2018.

352-353 Introduce this information in the methodology rather than in limitations.

  • All demographic information is referenced in the methodology, as is the location of environmental data capture. The seven-day delay to distribution of the survey has been added.

362-365 Very important for future research. To include questions in the survey concerning that issue specifically would be necessary.  

  • We agree that this is important and would likely require more in depth physiological investigation beyond that of a survey.

REFERENCES

Please, check all the references because the styles are mixed. You should report this section in Vancouver style.

  • Changed to Vancouver

Reviewer 2 Report

The article Performance and Thermal Perceptions of Runners Competing in the London Marathon: Impact of environmental conditions by Tim Vernon et al. is a mixed design study that analyzed historical data and found that the 2018 Virgin London Marathon weather was hotter than the previous London Marathons. The authors also found a positive correlation between maximum race day temperature and finish time for mass participants and championship runners, in which higher temperatures associate with slower finish times. This study provides evidence that temperature forecasting and thermal perception data could be valuable for marathon organizers in the design of countermeasures to avoid high temperatures, increase safety and implement appropriate cooling strategies. The manuscript is interesting, the methodology well described, and the results support the authors conclusions. Here are some minor observations to improve the manuscript.

Figure 3: Describe the statistical analysis that was performed in the figure legend. Figure 3a and 3b: Add an uppercase letter C for the oC in x-axis titles. Figure 3b:  This graph needs to be improved for clarity. There is an overlap between the y axis and the data points.

Line 29: Virgin Money London Marathon and its abbreviation VMLM are used interchangeably throughout the manuscript. However, for consistency, I recommend defining the abbreviation here when the name is first mentioned and only use VMLM after that. 

Line 35: particularlywhen -> add space; Line 44 and 45: oC -> superscript; Line 126: figure -> Figure; Line 163: 270minutes -> add space; Line 286: figure3b -> Figure 3b; Line 326: Roberts (2010) -> add appropriate reference, in this case is 18; Line 324: define WBGT abbreviation; Line 392: remove acknowledgement section if there are no acknowledgements. 

Author Response

Response to Reviewer Two

The article Performance and Thermal Perceptions of Runners Competing in the London Marathon: Impact of environmental conditions by Tim Vernon et al. is a mixed design study that analyzed historical data and found that the 2018 Virgin London Marathon weather was hotter than the previous London Marathons. The authors also found a positive correlation between maximum race day temperature and finish time for mass participants and championship runners, in which higher temperatures associate with slower finish times. This study provides evidence that temperature forecasting and thermal perception data could be valuable for marathon organizers in the design of countermeasures to avoid high temperatures, increase safety and implement appropriate cooling strategies. The manuscript is interesting, the methodology well described, and the results support the authors conclusions. Here are some minor observations to improve the manuscript.

  • Thank you for taking the time to review the manuscript and for your suggestions, please find below our response and the changes that we have made to the manuscript as a result.

Figure 3: Describe the statistical analysis that was performed in the figure legend. Figure 3a and 3b: Add an uppercase letter C for the oC in x-axis titles. Figure 3b:  This graph needs to be improved for clarity. There is an overlap between the y axis and the data points.

  • Amended

Line 29: Virgin Money London Marathon and its abbreviation VMLM are used interchangeably throughout the manuscript. However, for consistency, I recommend defining the abbreviation here when the name is first mentioned and only use VMLM after that. 

  • Done

Line 35: particularlywhen -> add space; Line 44 and 45: oC -> superscript; Line 126: figure -> Figure; Line 163: 270minutes -> add space; Line 286: figure3b -> Figure 3b; Line 326: Roberts (2010) -> add appropriate reference, in this case is 18; Line 324: define WBGT abbreviation; Line 392: remove acknowledgement section if there are no acknowledgements. 

  • All amended

Reviewer 3 Report

This article deals with an important and contemporary topic: heat and performance. This approach is not original especially in marathon, however, the authors tried an original approach linking pre-race and race temperatures and performances. Although their approach is interesting, several problems remained, the main one being the lack of clarity in the storyline of this study.

Major comment 1: The results presented could be separated in two. The first one is the impact of pre-race (60 days before the race) and the race temperatures on performance. This approach is partly original and interesting (see Major comment 3) but it appears that the analysis of the 2018 marathon is not very important. It only added a point in the analysis. The second one is the survey part. This approach may be somewhat limited (see Major comment 2) and associated with the temperature curves of 2018. I found that this second approach may be removed without decreasing the interest of the study.  However if the authors, the other reviewers or the editor find these results important to present, the manuscript’s storyline should be reordered in first presenting the 2018 marathon, its specific heat stress (temperature and survey results), and the induced decrease in performance then in putting this data in perspective and analyzing the global effect of temperature on performance. This order, if respected during the whole manuscript, should improve the manuscript clarity.

Major comment 2: the additional interest of the survey seems somewhat weak compared to the thermal analysis. First, the duration between the end of the race and the survey is intriguing especially if thermal sensation and comfort are assessed for each 10-km segment. Can the author explain this delay?

Major comment 3: the pre-race part is insufficiently introduced and requires more continuity along the manuscript. Indeed, this novelty sometimes seems to be a sub analysis of this study and sometimes (lines 238-239), it is spotlighted. For example, in the abstract, this analysis (“the difference in 16 maximum race day temperature and mean maximum daily temperature for the 60 days prior to the London Marathon (P < 0.05)”) came from nowhere.  Moreover, in the methods, we ignore whether the participants lives and/or trained in London in the previous 60 days and therefore faced the 2 extreme weather events. I guess that the authors have not access to the location of all the participants since 2001, but they are maybe aware of the location of the participants that filled the survey. This point is mentioned in the limitation part, but is should be extended. Moreover, in lines 269-270, the authors state that “This knowledge is important because it could be used before any London marathon to adjust pacing strategies by taking into account the St. James Park weather forecast”. This seems limited since the same recommendation may be done for runners that trained in other places and even for competition somewhere else.  

Major comment 4: Why relative humidity was not considered in this analysis? I checked using the met office website and these measurements are available at the Saint-James park location. Since humidity plays a major role in thermoregulation and the subjective thermal discomfort, this lack is very problematic unless the authors have a main reason. Using heat index or other metrics to take into account temperature and humidity would be interesting.

Major comment 5: The practical recommendations seem an over interpretation of the results. Point 1 concerns runners, trainers, and organizers. It should be compartmented. The importance of keeping detailed log of environmental temperature in the preparatory phases requires further explanation. Points 2 and 6 have no relation with the results. Points 4 and 5 are an over interpretation of the results. Point 3 seems justified.

Minor comments

Minor comment 1: Numerous double spaces (line 50), absence of space (line 6), and other mistakes (line 44, oC) are present all along the manuscript.

Minor comment 2, Lines 100-102: why the authors chose the maximum temperature values for the race on the 60 days before instead of a mean value? This should be justified.   

Minor comment 3, Lines 124 and 125: How did the authors manage to compare on single value (the mean temperature of the 218 marathon and the maximum daily temperature in the 60 days) to the remaining values of the previous years and 2019? The data analysis part is not helping.

Minor comment 4, Lines 285-295: The authors compared two performance decreases that are not comparable. In the first one, they used their regression line to predict that the decrease in performance should be 6.6% compared to “we do not clearly know what” conditions. In the second one, a 14% decrement in performance was notified between the actual performance of the survey participants and their estimated finish time implying that (if we forgot that this sample was not representative of all participants as stated by the authors) that they would have ran this marathon according to their estimation.  This comparison is therefore not justified and the following discussion not related.

Minor comment 5, Lines 288-295: It is strange that the authors did not extend the discussion on the difference of correlations between temperature and the running times of the two halves. It suggests that 1) participants are not able to anticipate the detrimental effects of heat on performance,  2) thermal sensation of the beginning of the marathon are not reliable, 3) decrease in performance occurred after mid-run and maybe other interesting hypotheses that could be discussed with references. Literature on the effect of heat on pacing is available and should be quoted.

Minor comment 6, Lines 325: In the Table 2 of the ref [21], they considered that high-risk individuals should not compete above 22.2 °C WBGT (cancellation of all competition > 27.8 °C WBGT). The authors used the columns for noncontinuous activities instead of continuous ones. Thus this recommendation is not very different from the one of Roberts (the reference is lacking).  

Author Response

Response to Reviewer Three

This article deals with an important and contemporary topic: heat and performance. This approach is not original especially in marathon, however, the authors tried an original approach linking pre-race and race temperatures and performances. Although their approach is interesting, several problems remained, the main one being the lack of clarity in the storyline of this study.

  • Thank you for taking the time to review the manuscript and for your suggestions, please find below our response and the changes that we have made to the manuscript as a result.

Major comment 1: The results presented could be separated in two. The first one is the impact of pre-race (60 days before the race) and the race temperatures on performance. This approach is partly original and interesting (see Major comment 3) but it appears that the analysis of the 2018 marathon is not very important. It only added a point in the analysis. The second one is the survey part. This approach may be somewhat limited (see Major comment 2) and associated with the temperature curves of 2018. I found that this second approach may be removed without decreasing the interest of the study.  However if the authors, the other reviewers or the editor find these results important to present, the manuscript’s storyline should be reordered in first presenting the 2018 marathon, its specific heat stress (temperature and survey results), and the induced decrease in performance then in putting this data in perspective and analyzing the global effect of temperature on performance. This order, if respected during the whole manuscript, should improve the manuscript clarity.

  • We have now reordered the results section to reflect your comments.
  • We feel that 2018 is important as this was the unique point in the analysis, with extreme weather events prior to and on the day of the marathon.
  • The survey data is important because it provides context to the extreme weather events of 2018 and forms the basis for our practical recommendations regarding preparation and response to marathon participation in unseasonably hot conditions.
  • We have switched the data around we do believe it has improved the clarity of the manuscript.

Major comment 2: the additional interest of the survey seems somewhat weak compared to the thermal analysis. First, the duration between the end of the race and the survey is intriguing especially if thermal sensation and comfort are assessed for each 10-km segment. Can the author explain this delay?

  • This piece of research was in response to the events on the day of the 2018 London Marathon, in paragraph 2 L36, we introduce the unique weather events of 2018 that prompted this investigation, hence the seven-day delay. We feel that this is an integral component of the investigation because it enabled the generation of the survey creation and collection of the data based upon this extreme weather event.  It is also the second aim of the paper (Line 59).

Major comment 3: the pre-race part is insufficiently introduced and requires more continuity along the manuscript. Indeed, this novelty sometimes seems to be a sub analysis of this study and sometimes (lines 238-239), it is spotlighted. For example, in the abstract, this analysis (“the difference in 16 maximum race day temperature and mean maximum daily temperature for the 60 days prior to the London Marathon (P < 0.05)”) came from nowhere.  Moreover, in the methods, we ignore whether the participants lives and/or trained in London in the previous 60 days and therefore faced the 2 extreme weather events. I guess that the authors have not access to the location of all the participants since 2001, but they are maybe aware of the location of the participants that filled the survey. This point is mentioned in the limitation part, but is should be extended. Moreover, in lines 269-270, the authors state that “This knowledge is important because it could be used before any London marathon to adjust pacing strategies by taking into account the St. James Park weather forecast”. This seems limited since the same recommendation may be done for runners that trained in other places and even for competition somewhere else.  

  • You are correct, the pre-race analysis is not the main focus of the paper and is there to provide context to the 2018 hot weather event. As we highlight in the manuscript this is because of the two extreme cold weather events that would likely impair preparation for UK based marathon participants and -posed thermo perceptual challenges during the unseasonably hot conditions of the 2018 marathon.  In the survey we compare perceptions of heat and thermal comfort within the training environment to the actual marathon, as you can see, there were clear differences. 
  • The participants in the London Marathon are predominantly UK based, and would more than likely have experienced the two extreme cold events that took place across Europe in the run up to the marathon.
  • While we acknowledge this limitation, we feel that it is minor and rather than extend this point further we would rather focus on the positives of the study.

Major comment 4: Why relative humidity was not considered in this analysis? I checked using the met office website and these measurements are available at the Saint-James park location. Since humidity plays a major role in thermoregulation and the subjective thermal discomfort, this lack is very problematic unless the authors have a main reason. Using heat index or other metrics to take into account temperature and humidity would be interesting.

  • In the discussion (L292), we do discuss relative humidity and the potential effects of changes in immeasurable micro-climate conditions caused by running in close proximity to others and in built up areas with limited air flow.
  • Prior to submission of the manuscript, understanding the importance of relative humidity as you rightly mention, we calculated Heat Index for all marathon years. However, there are several limitations, most importantly that the calculation of HI does not change the effective temperature in the ranges that we were dealing with and as such had no impact upon our analysis.  Furthermore, we felt that the practicality of HI is limited because, runners are less likely to understand it and would have to do additional calculations.  Whereas our approach is based upon temperature alone and is easily interpreted and acted upon.

Major comment 5: The practical recommendations seem an over interpretation of the results. Point 1 concerns runners, trainers, and organizers. It should be compartmented. The importance of keeping detailed log of environmental temperature in the preparatory phases requires further explanation. Points 2 and 6 have no relation with the results. Points 4 and 5 are an over interpretation of the results. Point 3 seems justified.

  • We have changed the heading to General practical recommendations, this section includes general practical recommendations as well as recommendations directly from our study that would help marathon runners and organisers to conduct events safely in hot environments. 
  • Communication of these strategies is important regardless of whether the data specifically comes from this investigation, and we believe that our data helps to put these general recommendations in context.

Minor comments

Minor comment 1: Numerous double spaces (line 50), absence of space (line 6), and other mistakes (line 44, oC) are present all along the manuscript.

  • Amended throughout, there are a few places where the justification looks like an additional space.

Minor comment 2, Lines 100-102: why the authors chose the maximum temperature values for the race on the 60 days before instead of a mean value? This should be justified.   

  • We chose maximum temperature to highlight the extreme nature of the weather events, when it comes to acclimation, in this case, a higher temperature during this training period would be better. Using the maximum temperature during this period reflects the “best case scenario” and further highlights how extreme 2018 was.

Minor comment 3, Lines 124 and 125: How did the authors manage to compare on single value (the mean temperature of the 218 marathon and the maximum daily temperature in the 60 days) to the remaining values of the previous years and 2019? The data analysis part is not helping.

  • We don’t understand this comment, we compared the maximum temperature on the day of the 2018 marathon with the maximum temperature on the day of all other London Marathons.

Minor comment 4, Lines 285-295: The authors compared two performance decreases that are not comparable. In the first one, they used their regression line to predict that the decrease in performance should be 6.6% compared to “we do not clearly know what” conditions. In the second one, a 14% decrement in performance was notified between the actual performance of the survey participants and their estimated finish time implying that (if we forgot that this sample was not representative of all participants as stated by the authors) that they would have ran this marathon according to their estimation.  This comparison is therefore not justified and the following discussion not related.

  • This data is comparable because we are comparing a prediction against an actual finish time and in the remaining part of this section, we explain why this discrepancy might have occurred.
  • In the comment below, you rightly highlight that participants might not be able to anticipate the detrimental effects of heat on performance which is highlighted in this section.

Minor comment 5, Lines 288-295: It is strange that the authors did not extend the discussion on the difference of correlations between temperature and the running times of the two halves. It suggests that 1) participants are not able to anticipate the detrimental effects of heat on performance,  2) thermal sensation of the beginning of the marathon are not reliable, 3) decrease in performance occurred after mid-run and maybe other interesting hypotheses that could be discussed with references. Literature on the effect of heat on pacing is available and should be quoted.

  • As this was not one of the main aims of the paper, we have removed this element of the analysis to avoid confusion.

Minor comment 6, Lines 325: In the Table 2 of the ref [21], they considered that high-risk individuals should not compete above 22.2 °C WBGT (cancellation of all competition > 27.8 °C WBGT). The authors used the columns for noncontinuous activities instead of continuous ones. Thus this recommendation is not very different from the one of Roberts (the reference is lacking).  

  • You are correct, we have amended the paper based on your comment.

Reviewer 4 Report

General comments:

The article investigated the impact of environmental conditions on Performance and Thermal Perceptions of Runners Competing in the London Marathon. This interesting topic has great applicability to 'real-world practice. It is clear that considerable efforts have been invested in this project. However, I think there are some improvements that could be made to enhance the linkages of certain points and to highlight the novelty factors further. The introduction requires some adjustments to inform on the physiological mechanisms mediating the performance decrement in extreme environments and the rationale for the outcome measures selected. Further, the authors must improve the discussion, I think that the mechanistic discussion is very weak if almost absent. The below comments are intended to explain this view and are hoped to be of benefit to the authors.

Specific comments

Introduction:

-L33-35: “Such demands are intensified in hot and humid conditions where finishing times are impaired [4-6], and in-race withdrawals increase, particularly when air temperature exceeds 20°C [6].” Could you please explain sufficiently the physiological mechanisms mediating the performance decrement in extreme environments

-L36-53: The authors reported interesting findings from the literature on the relationship between extreme weather and sports event organization but they did not explain sufficiently the underlying mechanisms of these changes, it is necessary to develop this part to clarify the backdrop of what is already known/not known and how these physiological and perceptual changes can influence our practice?

-L58-68: It is important to explain the rationale of this study, but the purpose needs to be more concise. How this analysis advances our knowledge about the thermal demands of extreme weather events? And how it can change our practice?

Methods:

-L70: I suggest that authors can use a protocol design to facilitate understanding of the used model by the readers

-L71: The authors collected survey data. However, they did not give details about the type of this data? What types of questions? On thermal comfort? On the perception of exertion? On well-being?

-L89-92: Did the authors perform analysis based on the frequency of participation in marathons?

-L118: Why did the authors use Mann Whitney as a post hoc test! Why they did not use other more robust tests like the Tukey HSD test or the Dunn-Bonferroni test?

Results:

 - The authors performed a global analysis involving all data from 2001 through 2019 but did not make a real comparison between years with different weather.

-The thermal sensations are shown in Figures 4 and 5, is this only the data of novice marathon runners. If not, please provide a subgroup analysis

- Why the authors did not perform multiple regressions to look for causal links between survey data (perception of temperature, and thermal comfort during the Marathon, expected finish time) and marathon performance.

Discussion:

-The mechanistic discussion is weak if almost absent. The authors just recalled and described their findings in the discussion part but they did not really discuss these results and they do not compare their results with what is observed in previous studies.

-Please consider including possible mechanisms explaining the performance decrement in hot weather and the relationship between thermal perception and marathon performance.

Author Response

Response to Reviewer Four

The article investigated the impact of environmental conditions on Performance and Thermal Perceptions of Runners Competing in the London Marathon. This interesting topic has great applicability to 'real-world practice. It is clear that considerable efforts have been invested in this project. However, I think there are some improvements that could be made to enhance the linkages of certain points and to highlight the novelty factors further. The introduction requires some adjustments to inform on the physiological mechanisms mediating the performance decrement in extreme environments and the rationale for the outcome measures selected. Further, the authors must improve the discussion, I think that the mechanistic discussion is very weak if almost absent. The below comments are intended to explain this view and are hoped to be of benefit to the authors.

  • Thank you for taking the time to review the manuscript and for your suggestions, please find below our response and the changes that we have made to the manuscript as a result.

Specific comments

Introduction:

-L33-35: “Such demands are intensified in hot and humid conditions where finishing times are impaired [4-6], and in-race withdrawals increase, particularly when air temperature exceeds 20°C [6].” Could you please explain sufficiently the physiological mechanisms mediating the performance decrement in extreme environments

  • We chose not to introduce the physiological mechanisms as this is not a paper that enables us to discus the physiology. Any comment here would be superficial based on the complex physiological mechanisms that contribute to performance impairment.

-L36-53: The authors reported interesting findings from the literature on the relationship between extreme weather and sports event organization but they did not explain sufficiently the underlying mechanisms of these changes, it is necessary to develop this part to clarify the backdrop of what is already known/not known and how these physiological and perceptual changes can influence our practice?

  • Please see above

-L58-68: It is important to explain the rationale of this study, but the purpose needs to be more concise. How this analysis advances our knowledge about the thermal demands of extreme weather events? And how it can change our practice?

  • We feel we have clearly explained the aims and rationale of the study as concise as possible, this paper is not about the thermal demands of extreme weather events.

Methods:

-L70: I suggest that authors can use a protocol design to facilitate understanding of the used model by the readers

  • We do not understand this comment as we are unsure what a “protocol design” refers to.

-L71: The authors collected survey data. However, they did not give details about the type of this data? What types of questions? On thermal comfort? On the perception of exertion? On well-being?

  • We will add the survey questions to the submission as supplementary material.

-L89-92: Did the authors perform analysis based on the frequency of participation in marathons?

  • Yes, please see line 183.
  • “For those who had run marathon's previously, the mean finish time for the 2018 VMLM was 40 ± 27 minutes slower than their previous best marathon time (317 ± 71 mins vs 278 ± 60 mins; P < 0.05). Out of 101 runners who had previously run a marathon, one runner ran faster in 2018 VMLM by approximately 1 minute.”

-L118: Why did the authors use Mann Whitney as a post hoc test! Why they did not use other more robust tests like the Tukey HSD test or the Dunn-Bonferroni test?

  • This has been changed and we used the Dunn-Bonferroni test instead, the result is the unchanged.

Results:

 - The authors performed a global analysis involving all data from 2001 through 2019 but did not make a real comparison between years with different weather.

  • We placed this data in context to the extreme weather events of 2018.

-The thermal sensations are shown in Figures 4 and 5, is this only the data of novice marathon runners. If not, please provide a subgroup analysis

  • Yes it is, this was from the survey of participants from 2018.

- Why the authors did not perform multiple regressions to look for causal links between survey data (perception of temperature, and thermal comfort during the Marathon, expected finish time) and marathon performance.

  • We feel that there is no practical application of doing this and do not feel this analysis would add to the paper. You would not be able to perform the calculation until either during or after the event and the equation we derive from our data in 2018 would not be generalisable to any other conditions.

Discussion:

-The mechanistic discussion is weak if almost absent. The authors just recalled and described their findings in the discussion part but they did not really discuss these results and they do not compare their results with what is observed in previous studies.

  • In the discussion we further interpret the results of our analysis and place this in context to findings our previous studies. Specifically, we highlight comparisons between performance impairment and previous studies and indicate the practical importance of our findings. We also discuss the potential reasons for the discrepancies between predicted marathon performance and actual marathon performance and the possible perceptual mechanisms that might play a key role in explaining these differences. Furthermore, we place our findings in context to previous studies that have suggested safe temperatures for marathon running and offer our opinion on how this might be influenced by environmental temperatures leading up to the event.

-Please consider including possible mechanisms explaining the performance decrement in hot weather and the relationship between thermal perception and marathon performance.

  • We don’t have any physiological data to make comparisons between this study and others, we reiterate this is not a mechanistic study and further research would be required to do so.

Round 2

Reviewer 1 Report

Thank you for taking into consideration the reviewer´s comments. It will be an important contribution to their field of knowledge.

Author Response

Thank you for your comments and your help in refining our paper.

Reviewer 3 Report

The authors modified some parts of the article based on the reviewers’ comments. Some modifications were rightly done but it appears very insufficient and several problems need to be addressed.

Major comment 1: For a study describing the effects of climate on performance, eluding the impact of humidity is quite problematic. The authors did discuss the role of humidity but it is very anecdotal. They indicated that they used humidity in using heat index. But this analysis was not conclusive and that it was more practical to use simple method for athletes. I totally agree with this but it should be indicated in the discussion.

Furthermore, from a practical point of view, I ignore whether the use of the temperature difference between the 60 days before race and the race is pertinent. In comparing the Figures 2a and 2b, we may have the impression that using this temperature difference is not more useful than just using the temperature during marathon to adapt marathon pacing. The whole analysis between lines 260 and 271 could be done using both methods indicating that the strategy to follow the temperature during the training period is not justified because it is way less practical that just waiting for the forecast the day if the marathon. Moreover, it is very difficult to know whether the two unusual cold episodes in the training period may have a particular impact on the performance the day of the marathon. It is way more likely that the reduction of performance would have been similar without these cold episodes. Heat acclimation could not be reached with training sessions below twenty degrees. So training between 0-10 °C or at 15-20°C (only during a few days) should produce the same training adaptations. This analysis should therefore be removed from this article or its importance should be deeply reduced.

Major comment 2:  I maintain the comment on recommendations. The only pertinent recommendation is to use forecast to adapt marathon pacing. It seems unjustified to proposer other recommendations that are not based on the articles results. This is the role of the reviews.

Major comment 3: Statistical analysis #1

First reviewing minor comment 3, Lines 124 and 125: How did the authors manage to compare on single value (the mean temperature of the 218 marathon and the maximum daily temperature in the 60 days) to the remaining values of the previous years and 2019? The data analysis part is not helping.

Authors’ response: We don’t understand this comment, we compared the maximum temperature on the day of the 2018 marathon with the maximum temperature on the day of all other London Marathons.

I was obviously not sufficiently clear. “The maximum daily temperature in the 60 days before the 2018 race day was also lower than previous years (11.3 ± 6.3°C vs. 12.1 ± 1.6°C; P < 0.01)”. The presence of p<0.01 indicates that the authors performed a statistical test. They apparently compared the maximum daily temperature in the 60 days before the 2018 race and before others London marathons. More details are required to understand this analysis: Did the authors compare the 60 maximum daily temperatures to the 60 temperatures of each other years (60 x 37 years = 2220 temperatures) ? In reading this part, we could understand that the mean maximal temperature during this 60 period (the “single value” in the previous comment) was compared to the 37 other mean values.  

Major comment 4: Statistical analysis #2

The use of the Friedman test to compare the thermal sensation and comfort data should be justified. Indeed, based on the survey, the authors did not obtain numeric values but rather a repartition of answers (from cold to very hot and from very uncomfortable to comfortable).  A comparison of frequencies (like Chi-square test) appeared more justified. The authors therefore need to bring more details on this analysis.

Major comment 5: Statistical analysis #3 (lines 281-283)

First draft minor comment 4, lines 285-295: The authors compared two performance decreases that are not comparable. In the first one, they used their regression line to predict that the decrease in performance should be 6.6% compared to “we do not clearly know what” conditions. In the second one, a 14% decrement in performance was notified between the actual performance of the survey participants and their estimated finish time implying that (if we forgot that this sample was not representative of all participants as stated by the authors) that they would have ran this marathon according to their estimation. This comparison is therefore not justified and the following discussion not related.

Authors’ response: This data is comparable because we are comparing a prediction against an actual finish time and in the remaining part of this section, we explain why this discrepancy might have occurred.

The authors’ response is not sufficient. I understand their response but the comparisons between these two decrement percentages is subjected to a major bias. We could expect that all participants in all the London Marathons regardless of temperature did not manage to respect their expected time (because it appears normal to be optimistic). Thus, it is possible that the difference between expected and actual performance was about 5-10% in all races meaning that the 14% and the 6.6% performance impairment are quite comparable.

Minor comment 1 (lines 79-80 and Figure 2c): I do not see the importance of differentiating Championship runners from the finishers. Since it was not present in the first draft, I assume it was a request from another reviewer. I understand that the authors modified the manuscript but it should be removed from the manuscript.  

Minor comment 2 (line 215): What means “body” in this sentence?

Minor comment 3 (line 282): the modification of the Figure’s number was not done.

Minor comment 4 (line 323): the word “respectively” is likely missing after the comma.

Author Response

Reviewer Three

The authors modified some parts of the article based on the reviewers’ comments. Some modifications were rightly done but it appears very insufficient and several problems need to be addressed.

Major comment 1: For a study describing the effects of climate on performance, eluding the impact of humidity is quite problematic. The authors did discuss the role of humidity but it is very anecdotal. They indicated that they used humidity in using heat index. But this analysis was not conclusive and that it was more practical to use simple method for athletes. I totally agree with this but it should be indicated in the discussion.

Author Response: We acknowledge the potential impact of humidity within the discussion, but we do not have data regarding the microclimate of runners. We have added the use of multiple environmental condition sensors into the future research section of the discussion to reflect the reviewers’ comments that relative humidity is an important component of human thermoregulation and that more data is required to elucidate the impact of humidity and mass participation marathon runner.

As we stated in our initial response before submission we investigation the use of the Heat Index. However, crucially the Heat Index calculated temperature is the same as ambient temperature up to 26°C and 50% RH. We did not observe temperatures that exceeded this threshold and therefore the heat index calculated temperature was exactly the same as the ambient temperature. For this reason, we did not include this in our analysis.

Furthermore, from a practical point of view, I ignore whether the use of the temperature difference between the 60 days before race and the race is pertinent. In comparing the Figures 2a and 2b, we may have the impression that using this temperature difference is not more useful than just using the temperature during marathon to adapt marathon pacing. The whole analysis between lines 260 and 271 could be done using both methods indicating that the strategy to follow the temperature during the training period is not justified because it is way less practical that just waiting for the forecast the day if the marathon.

Author response: The information regarding temperature 60 days before the marathon is important to the story of the manuscript and is placed in context by the quantification of thermal sensation and thermal comfort during training and during the marathon. As we state in the introduction:

“In the final weeks and days of marathon preparation runners were faced with two conflicting extreme weather events. In the future, extreme weather events are likely to increase in either frequency and / or severity, posing risks to the short and long-term health and wellbeing of marathon runners.”

We were interested in the effect of the 2 adverse cold weather events in the lead-up to the marathon and whether or not these impaired the preparation of runners when combined with the 3rd adverse weather event, a record hot marathon. As you correctly point out, using the temperature differential was not a better predictor of performance that using temperature alone.

Moreover, it is very difficult to know whether the two unusual cold episodes in the training period may have a particular impact on the performance the day of the marathon. It is way more likely that the reduction of performance would have been similar without these cold episodes. Heat acclimation could not be reached with training sessions below twenty degrees. So training between 0-10 °C or at 15-20°C (only during a few days) should produce the same training adaptations. This analysis should therefore be removed from this article or its importance should be deeply reduced.

We were interested in documenting thermal perception during training, when the weather was colder and then during the marathon, when the weather was hotter, rather than making comments regarding the potential for heat acclimation. However, we have updated the manuscript discussion to reflect your points here but feel the inclusion of the temperature differential analysis is still important to the narrative of the manuscript. 

Major comment 2:  I maintain the comment on recommendations. The only pertinent recommendation is to use forecast to adapt marathon pacing. It seems unjustified to proposer other recommendations that are not based on the articles results. This is the role of the reviews.

Author response: We have removed the additional recommendations based on your comment.

Major comment 3: Statistical analysis #1

First reviewing minor comment 3, Lines 124 and 125: How did the authors manage to compare on single value (the mean temperature of the 218 marathon and the maximum daily temperature in the 60 days) to the remaining values of the previous years and 2019? The data analysis part is not helping.

Authors’ response: We don’t understand this comment, we compared the maximum temperature on the day of the 2018 marathon with the maximum temperature on the day of all other London Marathons.

I was obviously not sufficiently clear. “The maximum daily temperature in the 60 days before the 2018 race day was also lower than previous years (11.3 ± 6.3°C vs. 12.1 ± 1.6°C; P < 0.01)”. The presence of p<0.01 indicates that the authors performed a statistical test. They apparently compared the maximum daily temperature in the 60 days before the 2018 race and before others London marathons. More details are required to understand this analysis: Did the authors compare the 60 maximum daily temperatures to the 60 temperatures of each other years (60 x 37 years = 2220 temperatures) ? In reading this part, we could understand that the mean maximal temperature during this 60 period (the “single value” in the previous comment) was compared to the 37 other mean values.  

Author response: We agree this part of the manuscript was confusing. We have removed the reference to the statistical analysis and left this as a simple comparison.

Major comment 4: Statistical analysis #2

The use of the Friedman test to compare the thermal sensation and comfort data should be justified. Indeed, based on the survey, the authors did not obtain numeric values but rather a repartition of answers (from cold to very hot and from very uncomfortable to comfortable).  A comparison of frequencies (like Chi-square test) appeared more justified. The authors therefore need to bring more details on this analysis.

Author response: The scales for thermal sensation and thermal comfort are numeric and ordinal and in the initial revised manuscript we provided the mean, standard deviation and the median of these values. They are anchored by qualitative descriptions but analysed as numerical values, as such a Friedman test is appropriate to analyse this data.

Major comment 5: Statistical analysis #3 (lines 281-283)

First draft minor comment 4, lines 285-295: The authors compared two performance decreases that are not comparable. In the first one, they used their regression line to predict that the decrease in performance should be 6.6% compared to “we do not clearly know what” conditions. In the second one, a 14% decrement in performance was notified between the actual performance of the survey participants and their estimated finish time implying that (if we forgot that this sample was not representative of all participants as stated by the authors) that they would have ran this marathon according to their estimation. This comparison is therefore not justified and the following discussion not related.

Authors’ response: This data is comparable because we are comparing a prediction against an actual finish time and in the remaining part of this section, we explain why this discrepancy might have occurred.

The authors’ response is not sufficient. I understand their response but the comparisons between these two decrement percentages is subjected to a major bias. We could expect that all participants in all the London Marathons regardless of temperature did not manage to respect their expected time (because it appears normal to be optimistic). Thus, it is possible that the difference between expected and actual performance was about 5-10% in all races meaning that the 14% and the 6.6% performance impairment are quite comparable.

Authors response: We have updated the manuscript to reflect the reviewers concerns here.

Minor comment 1 (lines 79-80 and Figure 2c): I do not see the importance of differentiating Championship runners from the finishers. Since it was not present in the first draft, I assume it was a request from another reviewer. I understand that the authors modified the manuscript but it should be removed from the manuscript.  

Author response: We have removed all references to championship runners based on the reviewers comments.

Minor comment 2 (line 215): What means “body” in this sentence?

Authors response: The body temperature of the runners, rather than the ambient temperature.

Minor comment 3 (line 282): the modification of the Figure’s number was not done.

Author response: Updated

Minor comment 4 (line 323): the word “respectively” is likely missing after the comma.

Author response: Updated

Reviewer 4 Report

The authors resubmitted their study focus on “ the impact of environmental conditions on Performance and Thermal Perceptions of Runners Competing in the London Marathon.” Overall, the authors did not take into consideration all the suggestions and did not follow the reviewer’s suggestions. I think that the manuscript still needs improvements to be to be published in IJERPH. The authors present their data in a superficial way without explaining the mechanisms responsible for the observed changes and the rationale of their manuscripts is not clear. 

Author Response

Reviewer Four

The authors resubmitted their study focus on “ the impact of environmental conditions on Performance and Thermal Perceptions of Runners Competing in the London Marathon.” Overall, the authors did not take into consideration all the suggestions and did not follow the reviewer’s suggestions. I think that the manuscript still needs improvements to be to be published in IJERPH. The authors present their data in a superficial way without explaining the mechanisms responsible for the observed changes and the rationale of their manuscripts is not clear.

Authors response: We acknowledged reviewer four’s comments in our initial replies. Crucially, this manuscript was not focussed on the physiological mechanisms regarding performance impairment due to the heat. This manuscript was an analysis of ambient temperature data, performance times and thermal perception. The reviewer must acknowledge that this is not a limitation of the manuscript because it was not the aim of the research to collect thermophysiological data of over 300 novices runners during a marathon, nor the >600,000 performance times within the analysis. As for the rationale of the manuscript this remains clear in the as per the initial submission – please note that none of the other reviewers had any issue with this aspect of the manuscript.

“the aims of this research were to 1) identify historical temperature data for all London Marathons to place the extreme weather events of 2018 in context; 2) survey a sample of novice-mass participation marathoners who started the 2018 Virgin Money London Marathon VMLM to examine the thermal demands of the extreme weather event on race day and 3) investigate the effect of the extreme weather events on finish time in both mass participation and championship runners. This investigation is the first in the context of the London marathon and warranted on the basis of 1) the addition to our knowledge and understanding of perceptual demands of mass participation runners in the Virgin Money London Marathon, and 2) identification of potential areas whereby participants and race organisers might seek to change their practice in preparation for, or on the day of the event when high-temperatures are forecast.”